# Dual Crop Coefficient Approach in *Vitis vinifera* L. cv. Loureiro

Simão P. Silva [1,*], M. Isabel Valín [1], Susana Mendes [1], Claúdio Araujo-Paredes [2] and Javier J. Cancela [3]

1   Centre for Research and Development in Agrifood Systems and Sustainability (CISAS), Escola Superior Agrária, Instituto Politécnico de Viana do Castelo, 4900-347 Viana do Castelo, Portugal; isabelvalin@esa.ipvc.pt (M.I.V.); smendes@esa.ipvc.pt (S.M.)
2   Research Unit in Materials, Energy and Environment for Sustainability (PROMETHEUS), Escola Superior Agrária, Instituto Politécnico de Viana do Castelo, 4900-347 Viana do Castelo, Portugal; cparedes@esa.ipvc.pt
3   GI-1716 Projects and Planification, Agroforestry Engineering Department, Escuela Politécnica Superior de Ingeniería Lugo, University of Santiago de Compostela, 27002 Lugo, Spain; javierjose.cancela@usc.es
*   Correspondence: silvasimao@esa.ipvc.pt

**Abstract:** Vineyard irrigation management in temperate zones requires knowledge of the crop water requirements, especially in the context of climate change. The main objective of this work was to estimate the crop evapotranspiration ($ET_c$) of *Vitis vinifera* cv. Loureiro for local conditions, applying the dual crop coefficient approach. The study was carried out in a vineyard during two growing seasons (2019–2020). Three irrigation treatments, full irrigation (FI), deficit irrigation (DI), and rainfed (R), were considered. The $ET_c$ was estimated using the SIMDualKc model, which performs the soil water balance with the dual Kc approach. This balance was performed by calculating the basal coefficients for the grapevine ($K_{cb\ crop}$) and the active soil ground cover ($K_{cb\ gcover}$), which represent the transpiration component of $ET_c$ and the soil evaporation coefficient ($K_e$). The model was calibrated and validated by comparing the simulated soil water content (SWC) with the soil water content data measured with frequency domain reflectometry (FDR). A suitable adjustment between the simulated and observed SWC was obtained for the 2019 R strategy when the model was calibrated. As for the vine crop, the best fit was obtained for $K_{cb\ full\ ini}$ = 0.33, $K_{cb\ full\ mid}$ = 0.684, and $K_{cb\ full\ end}$ = 0.54. In this sense, the irrigation schedule must adjust these coefficients to local conditions to achieve economically and environmentally sustainable production.

**Keywords:** active soil ground cover; soil water content; soil evaporation; vine plants

## 1. Introduction

The sustainability of wine production is a global strategy that includes all stages of the wine production cycle. In achieving sustainable wine production, the environmental aspects, such as efficient water use, should be considered [1]. Water is a factor that significantly influences crop yields [2], and it must be applied sustainably and efficiently [3]. This application must be efficient because, according to predictions, global water extraction will increase by 55% between 2000 and 2050 in a similar scenario to the current one [4]. In addition, predictions estimate a 20% decrease in precipitation in the *Vinhos Verdes* region, according to the IPCC RCP8.5 scenario [5].

To implement a sustainable strategy, it is necessary to expand the knowledge on crop evapotranspiration to support appropriate irrigation scheduling and management [5]. The estimation of crop evapotranspiration can be obtained with the single or dual crop coefficient approach (dual Kc) [6] through various field measurements techniques, for example, remote sensor data [7,8], eddy covariance [9], lysimeter [10,11], and measurements of soil water content [12,13].

The dual Kc approach estimates the actual crop evapotranspiration ($ET_{c\ act}$) through reference evapotranspiration ($ET_o$) and actual crop coefficient ($K_{c\ act}$). Adopting this dual Kc approach (Equation (1)), the $K_{c\ act}$ results from the sum of the basal crop coefficients

($K_{cb}$), correcting through a stress coefficient ($K_s$) that varies between 0 (maximum stress) and 1 (no stress), with the soil evaporation ($K_e$) [6]. Thus, $ET_{c\,act}$ is defined as follows:

$$ET_{c\,act} = (K_s \times K_{cb} + K_e) \times ET_o = K_{c\,act} \times ET_o \tag{1}$$

where $K_s$ represents the stress coefficient, $K_{cb}$ is the basal crop coefficient, $K_e$ is the soil evaporation coefficient, and $ET_o$ represents the reference evapotranspiration (mm d$^{-1}$).

During the active growing season, variations in the density and height of active soil ground cover are observed in the vineyards of this region. These variations are influenced by cultural operations aimed at controlling the active soil ground cover's vegetative vigor (herbicide application or tillage operations), as well as by climatic factors [14]. The water consumption of the active soil ground cover is environmentally acceptable when considering the numerous ecosystem services provided by cover crops in vineyards [15], such as the prevention of erosion [16], the contribution to carbon (C) sequestration, the increase in total nitrogen (N) content [17], and the contributions to plant vigor and fruit quality control [18], as well as pest control [19]. However, water consumption must also be considered in crop evapotranspiration [9]. In this sense, following the dual Kc approach, the active soil ground cover represents a part of the evapotranspiration. Therefore, the total evapotranspiration results from the sum of the transpiration from active soil ground cover and vineyard with the soil evaporation [20].

The dual Kc approach for computing and partitioning crop evapotranspiration can be facilitated using index and models applications, such as the Satellite-based NDVI [21], AquaCrop model [22], and SIMDualKc model [23,24]. The SIMDualKc performs the soil water balance at the field level, using a daily timestep, based on Equation (2):

$$D_i = D_{i-1} - (P - RO)_i - I_i - CR_i + ET_{c,\,i} + DP_i \tag{2}$$

where $D_{i-1}$ is the depletion at the preceding day, P is the precipitation, I is the irrigation, CR is the capillary rise from a shallow water table, DP is the deep percolation out of the root zone, $ET_{c_i}$ is the crop evapotranspiration (mm), and RO is the runoff, with all terms expressed in mm.

This model was validated using data from several permanent crops with active soil ground cover, namely olives [8,9] and hop [25]. The model was also validated for the culture of *Vitis vinifera*, where studies on cv. Albariño [20] and Godello e Mencía [26], took into account the effect of active ground cover. In these studies, the SIMDualKc model has been shown to be appropriate for adopting the dual Kc approach. In addition, to help computing and partitioning crop evapotranspiration, SIMDualKc has been used to update dual crop coefficients for many crops [27–31].

The present study aims to provide information on evapotranspiration and crop coefficients for *Vitis vinifera* cv. Loureiro in northwest Portugal. The specific objectives of this paper are: (a) to calibrate and validate the SIMDualKc model using the dual Kc approach, (b) to provide the $K_{cb}$ for each growth stage of the crop, (c) to apply the dual Kc approach to different irrigation strategies and (d) to verify if cv. Loureiro cultivated without irrigation (the most practiced in the region) is cultivated under drought stress. The results obtained in this work aim to support irrigation management programs, improving the use of water in agriculture.

## 2. Materials and Methods

### 2.1. Study Area

The study was carried out in a commercial cv. Loureiro vineyard, located in Ponte de Lima, in northwest Portugal (41°40′32″N, 8°32′6″ W, and 170 m.a.s.l.) during two growing seasons (2019 and 2020). The vineyard was planted in 2001, with a distance between rows of 3.0 m and a distance of 2.0 m between vines (1666 plant ha$^{-1}$). The vineyard is north–south orientated and trained to a single upward cordon. Drip irrigation was installed as the irrigation system, with one drip line per row, located at 40 cm above the

ground and pressure-compensated emitters (4 L h$^{-1}$) separated by one meter. The climate is Atlantic, characterized by relatively high annual rainfall (above 1200 mm) and relatively mild summers [32]. The Köppen–Geiger [33] classification is Csb.

The daily meteorological data (temperature, humidity, wind speed at 2 m height, net radiation, and precipitation) were collected from the weather station (UNL Ameriflux Site, Mead NE) located in the field. The ET$_o$ was computed with the FAO Penman–Monteith Equation (3) [6], which, for the computation of daily timesteps, takes the following form:

$$ET_o = \frac{0.408\,\Delta(R_n\,-\,G) + \gamma\,\frac{900}{T+273}U_2(e_a\,-\,e_d)}{\Delta + \gamma\,(1\,-\,0.34U_2)} \tag{3}$$

where $\Delta$ represents the slope of the saturation vapor pressure-temperature relationship at the mean air temperature (kPa °C$^{-1}$), $R_n$ is the net radiation at the crop surface (MJ m$^{-2}$ d$^{-1}$), G is the soil heat flux density (MJ m$^{-2}$ d$^{-1}$), $\gamma$ is the psychometric constant (kPa °C$^{-1}$), T is the mean daily air temperature (°C), $U_2$ is the wind speed (m s$^{-1}$) at 2 m height and ($e_a\,-\,e_d$) represents the vapor pressure at the reference height of 2 m and deficit of air (kPa) at 2 m height.

The total rainfall was 1431 and 1590 mm in 2019 and 2020, respectively. In the growing season, the rainfall was 402 mm and 321 mm for 2019 and 2020, respectively (Figure 1). The ET$_o$ was higher in the summer, reaching 116 mm month$^{-1}$ in May 2019 and 126 mm month$^{-1}$ in July 2020 (Figure 1). However, its daily variability was high depending on the net radiation, temperature, and wind speed.

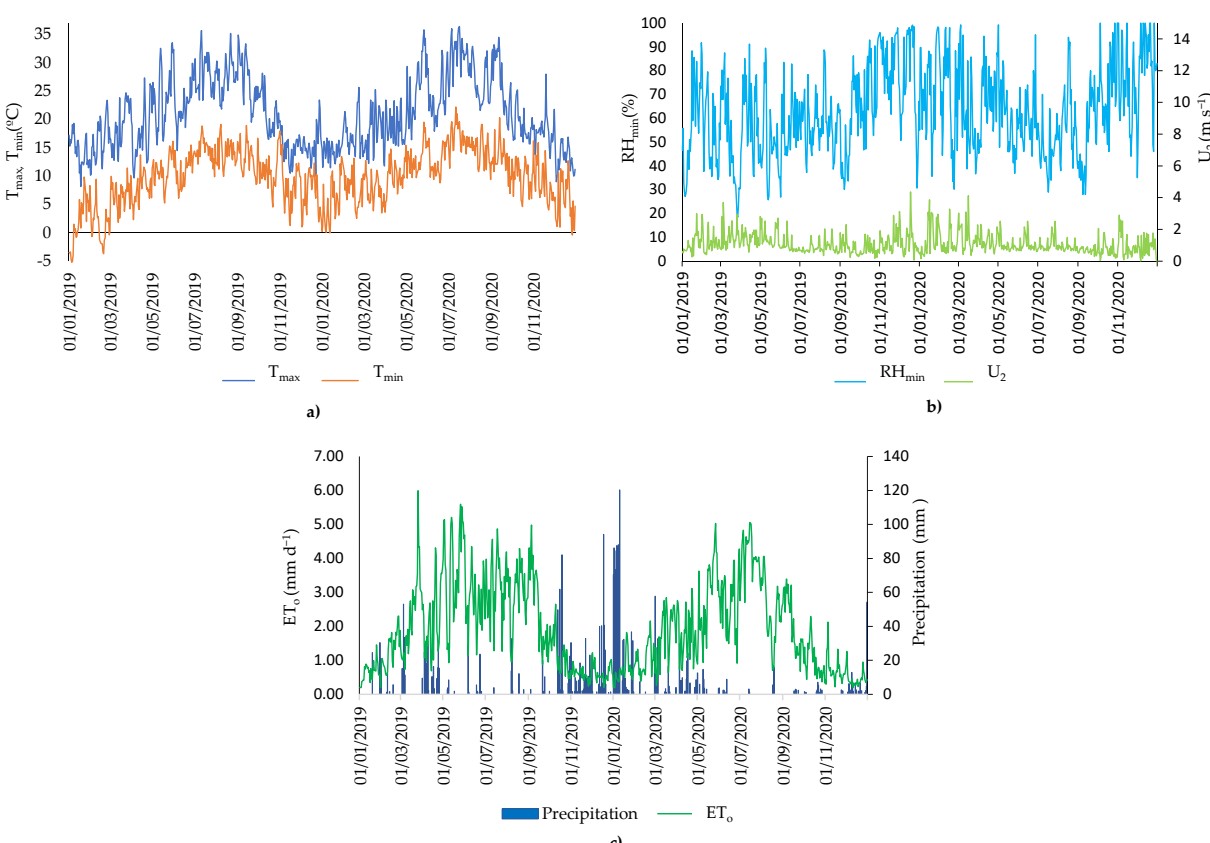

**Figure 1.** Daily weather data recorded in the field relative to (**a**) maximum (T$_{max}$, °C) and minimum (T$_{min}$, °C) temperatures; (**b**) minimum relative humidity (RH$_{min}$, %), wind speed at 2 m height (U$_2$, m s$^{-1}$) and (**c**) precipitation and reference evapotranspiration (ET$_o$, mm d$^{-1}$) for the period 2019–2020.

## 2.2. Climate and Soil Characterization

The climate directly and indirectly affects crop production and bioclimatic indices can be used to characterize the climate. Therefore, in this work, we used the Winkler index (WI), Huglin index (HI), Cool night index (CI), Seljaninov index (SI), and Branas index (BI), which have already been used by other authors in this region [34]. According to the climatic characterization shown in Figure 1, the growing seasons were very different with respect to the temperatures and precipitation values observed. The results of applying Equations (4) and (5) in 2019 are lower than the results for 2020 (Table 1). Thus, it is possible to verify that, in general, lower temperatures were observed in 2019. However, according to Equation (6), the minimum temperature in September 2019 was higher than the minimum temperature recorded in the same period in 2020 (Table 1). As the precipitation values were very different (Figure 1), the results of the application indexes that relate temperatures with precipitation Equations (7) and (8) were also very different in the two years of the study. Therefore, the results of Equations (7) and (8) obtained for 2019 were different from those obtained for 2020(Table 1).

**Table 1.** List of the bioclimatic indices used for this study for both years, their corresponding mathematical definitions, and sources.

| Index and Abbreviation | Equation | | Source | Results | |
|---|---|---|---|---|---|
| | | | | 2019 | 2020 |
| Winkler index (WI) | $\sum\limits_{01.04}^{30.10} (T_{avg} - 10\,°C)$ | (4) | [35] | 1771 | 2124 |
| Huglin index (HI) | $\sum\limits_{01.04}^{30.09} \frac{(T_{avg}-10)+(T_{max}-10)}{2} \times k$ | (5) | [36] | 2847 | 2937 |
| Cool night index (CI) | $T_{min}\text{average (september)}$ | (6) | [37] | 13.22 | 13.05 |
| Seljaninov index (SI) | $\sum(P/(\sum((T_{avg} - 10\,°C))))$ | (7) | [38] | 0.27 | 0.29 |
| Branas index (BI) | $\sum\limits_{01.04}^{31.08} T_{avg} \times P_{month}$ | (8) | [39] | 18,654 | 11,677 |

$T_{máx}$ is maximum air temperature (°C), $T_{min}$ is minimum air temperature (°C), $T_{avg}$ is mean air temperature (°C), $k$ is the length of day coefficient, and $P$ is precipitation (mm).

The soil was classified as eutric regosol [40], with a sandy loam texture. On average, the soil contained 70.2% sand, 20.3% silt, and 9.5% clay, with 1.95% organic matter. The total available soil water (TAW) at a depth of 0.8 m (m) was 100 mm. The TAW was calculated from the difference between the soil field capacity and the permanent wilting point using eight soil layers.

The field capacity and wilting point were obtained in the laboratory. To determine the field capacity, a pressure plate was used to apply a suction of $-1/3$ of the atmosphere to the saturated soil samples. With the same samples, a suction of $-15$ atmospheres was applied to determine the wilting point [41]. The field capacity and wilting point values were 0.251 and 0.121 $cm^3\,cm^{-3}$, respectively. Capacitive probes were used to determine the field capacity value in situ in eight different soil layers up to 0.8 m, which ranged between 0.197 and 0.273 $cm^3\,cm^{-3}$ for the depths of 0.1 m and 0.8 m, respectively. These values were used in the SWC simulation.

## 2.3. Experimental Design: Data Required to Apply SIMDualKc

Three irrigation treatments were considered, full irrigation (defined by the vinegrower; FI), deficit irrigation (DI), and rainfed (R), which is most practiced in the region. Each treatment had two replicates (Figure 2a) with four rows; in the two central rows were installed access probes and the other two rows were used as buffers (Figure 2a). To obtain the SWC readings, 24 access probe tubes (80 cm depth) were installed in the rows (Figure 2b), distributed over the treatments (8 access probe tubes per treatment). In 2019, irrigation was implemented between DOY 142 and DOY 238 to FI treatment when the volumetric soil moisture content was 90% of field capacity and reached 69.67 mm (10 irrigation events),

while for the treatment DI was implemented between DOY 204 and DOY 217 when the volumetric soil moisture content was 70% of field capacity and reached 17.33 mm (two irrigation events). In 2020, the irrigation was implemented between DOY 181 and DOY 224, when the volumetric soil moisture content was 70% of field capacity and reached 94 mm (10 irrigation events) in FI, whereas the DI reached 32 mm (6 irrigation events), FI in 2020 started later than in 2019 due to problems in the irrigation system.

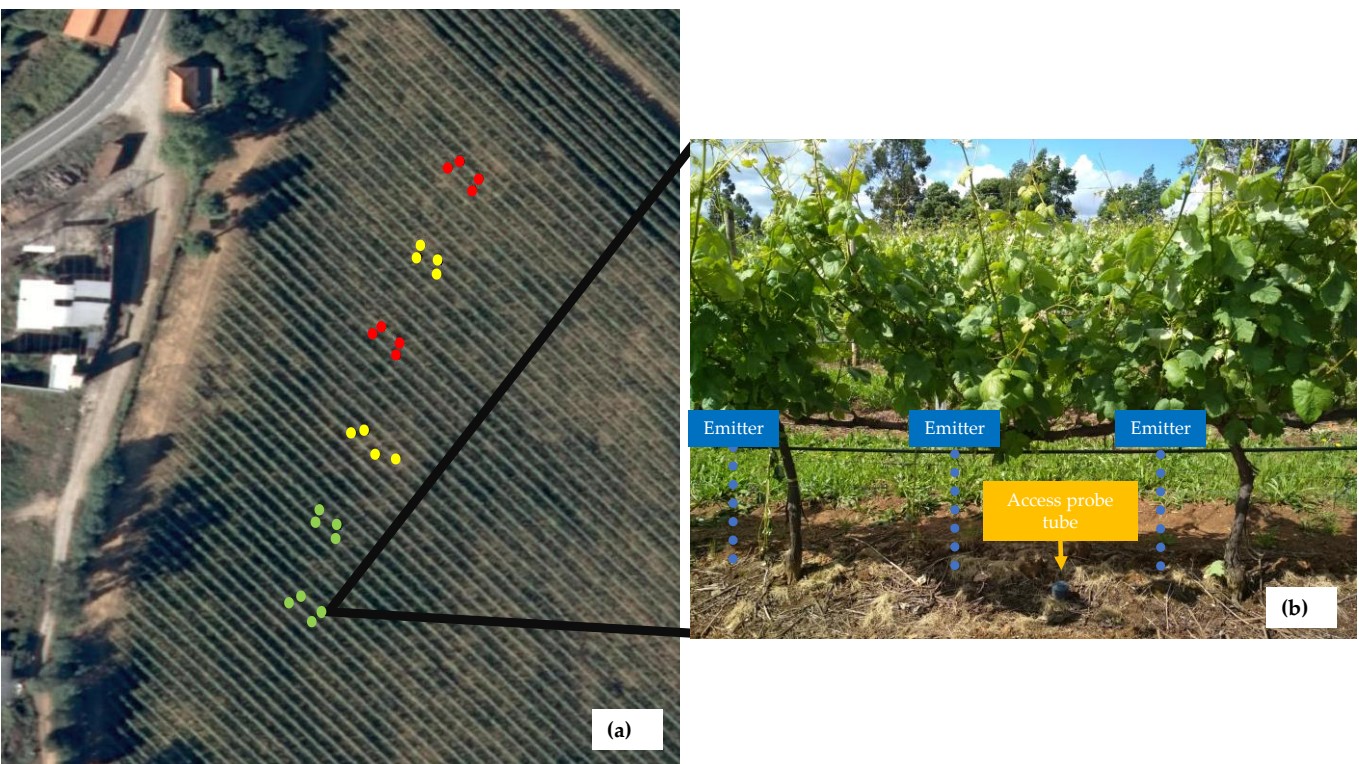

**Figure 2.** Position of the access probe tubes located in full irrigation (●), deficit irrigation (●), and rain-feed (●) treatments (**a**). A schematic view of vines, emitters, and access probe tubes (**b**).

Based on the successive SWC readings carried out with a capacitive probe (Diviner 2000) after the occurrence of high precipitation, the drainage depletion curve for the soil was obtained by applying the power regression method, Equation (9), defined by Liu et al. (2006) [42]:

$$SWC = a \times t^b \tag{9}$$

where *SWC* represents soil water content in mm, *a* is soil water storage value comprised between field capacity and saturation of the soil, *b* represents the velocity of drainage, and *t* is the time in days. The parameters *a* and *b* were defined as 287.05 and −0.056, respectively. A curve number of 60 was used to calculate the surface runoff [23,43].

Phenological evolution was performed based on the Baggiolini scale [44], which was later adapted to our FAO standard crop growth stages. Therefore, the day when the vine reached stage D (leaf emergence) corresponds to the start of the rapid growth stage [6], and the midseason stage corresponds to the period between phases: I (flowering) and phase M (veraison).

Crop heights (h) (from the soil) and the fraction of soil shaded by the crop ($f_c$), was determined through visual observation at solar noon and were observed throughout the active growing season. The values at the dates of the crop growth stages are presented in Table 2.

**Table 2.** Vineyard crop growth stages, height (h), and fraction of the soil covered by the crop in the vineyard ($f_c$).

| Crop Growth Stages | 2019 | | | 2020 | | |
|---|---|---|---|---|---|---|
| | Dates | H (m) | $f_c$ | Dates | H (m) | $f_c$ |
| Initiation | 21 March | 1.0 | 0.01 | 1 March | 1.0 | 0.01 |
| Start rapid growth | 26 March | 1.4 | 0.05 | 28 March | 1.2 | 0.05 |
| Start midseason | 6 June | 2.0 | 0.30 | 22 June | 2.0 | 0.30 |
| Start maturity | 2 September | 2.4 | 0.40 | 20 August | 2.4 | 0.40 |
| Harvesting | 23 September | 2.4 | 0.40 | 9 September | 2.4 | 0.40 |

The active ground cover was composed of spontaneous species. In 2019, the density of the active coverage (performed by observation) varied between 10% and 90%, with height varied between 0.02 and 0.25 m between the rows and no active coverage in the row (Table A1, Figure A1). In 2020, the density of active soil cover in the row ranged between 5% and 15%, and the height ranged between 0.05 and 0.10 m. Between the rows, the density ranged from 20% to 85%, and the height ranged between 0.05 and 0.30 m.

*2.4. Model Calibration and Validation*

The calibration of the SIMDualKc model [20,43,45] was performed by adjusting the crop parameters, basal crop coefficient initial ($K_{cb\ full\ ini}$), basal crop coefficient midseason ($K_{cb\ full\ mid}$), basal crop coefficient end ($K_{cb\ full\ end}$), and p depletion fractions, the soil evaporation parameters, depth of the evaporable layer ($Z_e$), total evaporable water (TEW) readily evaporable water (REW), and the local conditions of the cv. Loureiro by minimizing the residual deviations between the simulated and observed soil water content [26]. The calibration was performed by minimizing the differences between the observed and simulated SWC values in the treatment R for the year 2019. The validation used previously calibrated parameters in other treatments during 2019 and 2020. After calibration, the values of $K_{cb\ gcover}$, $K_{cb\ crop}$, $K_{cb\ (gcover+crop)\ act}$ (basal crop and active ground cover coefficient adjusted to climate and actual conditions), and $K_e$ could be determined to study the influence of active ground cover and vineyard, in transpiration and soil evaporation processes.

The procedures to assess the goodness-of-fit of the model were similar to those adopted in previous studies [20,43,45]. Linear regression was performed between the observed and simulated SWC values forced to the origin. A set of goodness-of-fit indicators was used to assess model fitting during the calibration and to evaluate the validation results. The indicators used were the regression coefficient (b), determination coefficient ($r^2$), root mean square error (RMSE, mm) Equation (10), normalized RMSE (NRMSE, %), average relative error (ARE, %) Equation (11), percent bias of estimation (PBIAS, %) Equation (12), modeling efficiency (EF, dimensionless) Equation (13), average absolute error (AAE, mm) Equation (14) and index of agreement ($d_{IA}$, dimensionless) Equation (15) between the observed and model-predicted values, respectively ($O_i$ and $P_i$ (i = 1, 2, ... , n)).

$$\text{RMSE} = \left[ \frac{\sum_{i=1}^{n} (P_i - O_i)^2}{n} \right]^{0.5} \tag{10}$$

$$\text{ARE} = \frac{100}{n} \sum_{i=1}^{n} \left| \frac{O_i - P_i}{O_i} \right| \tag{11}$$

$$\text{PBIAS} = 100 \frac{\sum_{i=1}^{n} (O_i - P_i)}{\sum_{i=1}^{n} (O_i)} \tag{12}$$

$$\text{EF} = 1.0 - \frac{\sum_{i=1}^{n} (O_i - P_i)^2}{\sum_{i=1}^{n} (O_i - \overline{O})^2} \tag{13}$$

$$AAE = \frac{1}{n} \sum_{i=1}^{n} |O_i - P_i| \tag{14}$$

$$dIA = 1.0 - \frac{\sum_{i=1}^{n} (P_i - O_i)^2}{\sum_{i=1}^{n} \left( |P_i - \overline{O}| + |O_i - \overline{O}| \right)^2} \tag{15}$$

## 3. Results

The standard and calibrated crop parameters ($K_{cb\ full\ ini}$, $K_{cb\ full\ mid}$, and $K_{cb\ full\ end}$, p) are presented in Table 3 [6,29]. $K_{cb\ full\ ini}$ was slightly higher than the standard values, while in the case of $K_{cb\ full\ mid}$ and $_{Kcb\ end}$, the values were similar to those proposed by the authors of [29]. In addition, there was a higher mean *p*-value than the reference.

**Table 3.** Standard and calibrated model parameters.

| Parameters | Standard | Source | Calibrated |
|---|---|---|---|
| $K_{cb\ full\ ini}$ (dimensionless) | 0.20 | | 0.33 |
| $K_{cb\ full\ mid}$ (dimensionless) | 0.80 | [29] | 0.684 |
| $K_{cb\ full\ end}$ (dimensionless) | 0.60 | | 0.54 |
| $p_{ini}$ (dimensionless) | 0.45 | | 0.45 |
| $p_{mid}$ (dimensionless) | 0.45 | [6] | 0.54 |
| $p_{end}$ (dimensionless) | 0.45 | | 0.45 |

$K_{cb}$ = basal crop coefficients, p = depletion fraction.

The variations of the different coefficients $K_e$, $K_{cb\ full}$, and $K_{c\ act}$ are shown in Figure 3, along with the values of precipitation and irrigations. In Figure 3, for each treatment, the stress level ($K_s$) varied between 0 (maximum stress) and 1 (no stress). In the R and DI treatments, values lower than 1 were obtained in both seasons.

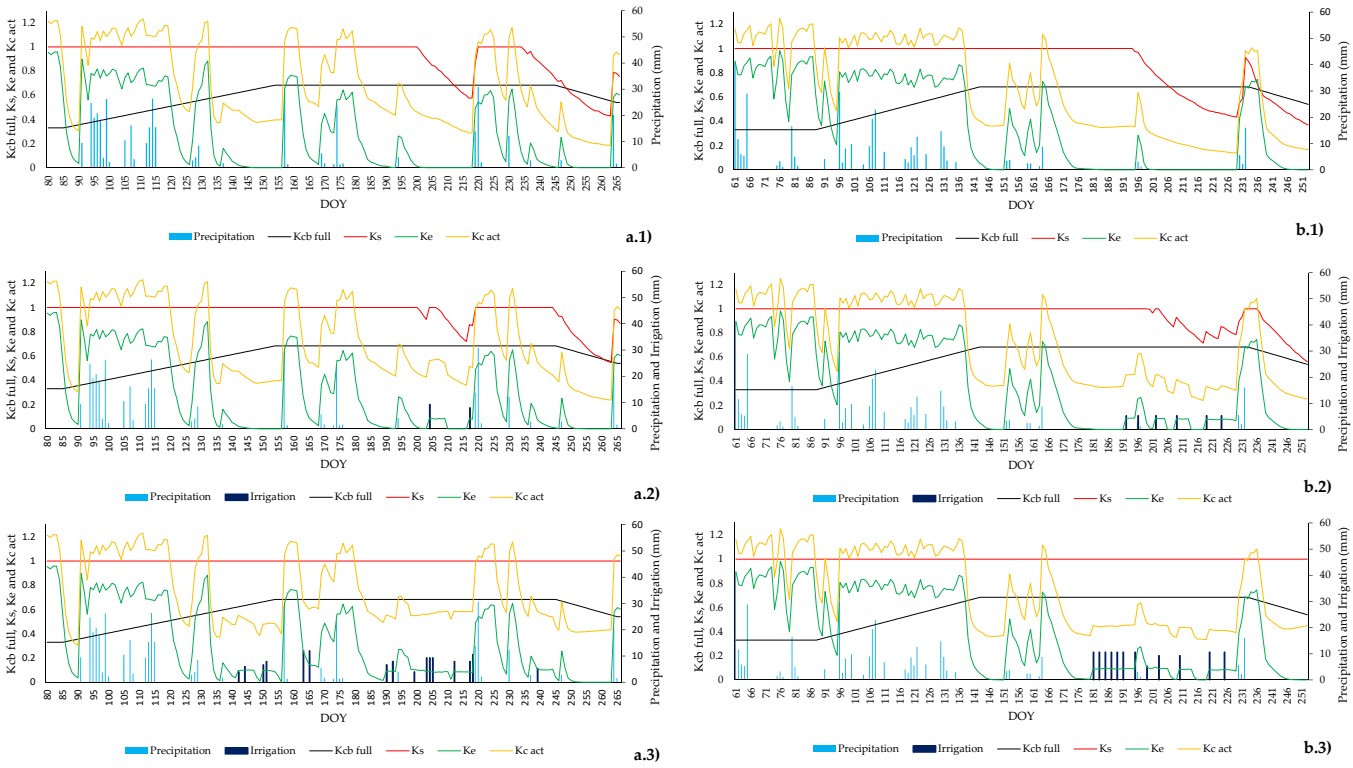

**Figure 3.** Precipitation (▬), irrigation (▬) $K_{cb\ full}$ (──), $K_s$ (──), $K_e$ (──), and $K_{c\ act}$ (──) and relative to the: (**a**) year 2019, (**b**) year 2020; (1) rainfed (R), (2), deficit irrigation (DI), (3) full irrigation (FI).

The total evaporable water (TEW) used was 17 mm, while the readily evaporable water (REW) used was 10 mm, and the depth of the soil evaporation layer ($Z_e$) was 0.1 m.

The effect of ground cover variations (Table A1) in $K_{cb\ gcover}$ was negligible and limited to values 0.15–0.27 for both seasons (Table 4). Annual climate conditions modify the $K_{cb\ crop}$, mainly during midseason and end season, with higher values in 2019 than in 2020. In the case of $K_{cb\ (gcover+crop)\ act}$ and $K_e$, similar values were achieved during initial and rapid growth growing seasons for both study years (2019 and 2020). To $K_{cb\ (gcover+crop)\ act}$, higher values were determined to FI, with respect to DI, where R treatment showed lower values, to mid and end season. The average $K_e$ values to R treatment were lower than DI and FI treatments during midseason, with slight differences during the end season. The values of $K_e$ during the crop growing stages are directly proportional to SWC.

**Table 4.** Average values of crop coefficients (-) and precipitation (mm) for different crop growth stages.

| Crop Growth Stages | | 2019 | | | | | | | | |
|---|---|---|---|---|---|---|---|---|---|---|
| | Dates | $K_{cb\ gcover}$ | $K_{cb\ crop}$ | $K_{cb\ (gcover+crop)\ act}$ | | | $K_e$ | | | Prec. (mm) |
| | | | | R | DI | FI | R | DI | FI | |
| Initial | 21 May/25 May | 0.26 | 0.00 | 0.26 | 0.26 | 0.26 | 0.93 | 0.93 | 0.93 | 0 |
| Rapid growth | 26 May/5 June | 0.24 | 0.13 | 0.37 | 0.37 | 0.37 | 0.40 | 0.40 | 0.42 | 238 |
| Midseason | 6 June/1 September | 0.27 | 0.21 | 0.45 | 0.47 | 0.48 | 0.22 | 0.23 | 0.27 | 131 |
| End season | 2 September/23 September | 0.15 | 0.27 | 0.25 | 0.31 | 0.42 | 0.10 | 0.10 | 0.10 | 25 |
| | | **2020** | | | | | | | | |
| | Dates | $K_{cb\ gcover}$ | $K_{cb\ crop}$ | $K_{cb\ (gcover+crop)\ act}$ | | | $K_e$ | | | Prec. (mm) |
| | | | | R | DI | FI | R | DI | FI | |
| Initial | 1 Mar/27 May | 0.27 | 0.00 | 0.27 | 0.27 | 0.27 | 0.83 | 0.83 | 0.83 | 122 |
| Rapid growth | 28 May/21 June | 0.25 | 0.09 | 0.33 | 0.33 | 0.33 | 0.52 | 0.52 | 0.52 | 206 |
| Midseason | 22 June/19 August | 0.20 | 0.15 | 0.27 | 0.32 | 0.35 | 0.04 | 0.08 | 0.10 | 28 |
| End season | 20 August/9 September | 0.17 | 0.22 | 0.21 | 0.30 | 0.39 | 0.19 | 0.20 | 0.21 | 0 |
| Average 2019–2020 | | 0.23 | 0.13 | 0.30 | 0.38 | 0.40 | 0.40 | 0.41 | 0.42 | 375.1 |

$K_{cb\ gcover}$ = basal crop coefficient of ground cover, $K_{cb\ crop}$ = basal crop coefficient of vineyard, $K_{cb\ (gcover+crop)\ act}$ = actual basal crop coefficient, $K_e$ = soil evaporation coefficient, Prec. = precipitation (mm). R = rainfed, DI = deficit irrigation, and FI = full irrigation treatments.

The results of the observed and simulated SWC values in R 2019, as well as the validation data, can be seen in Figure 4.

In Figure 4, the RAW value was 45 mm at the initial stage, however as the crop approached the midseason, this value increased progressively until reaching 54 mm, remaining stable until the moment of the beginning of maturation, then, it decreased again to 45 mm at the final stage.

The results of the set of goodness-of-fit indicators used to assess model fitting during calibration and to evaluate the validation results are presented in Table 5. In Table 5, the column of the underlined values refers to the calibration performed for the R-2019 treatment, as well as the validations for the other treatments. The b and $r^2$ values resulting from a forced regression to the origin can be observed, and the values of the indicators EF, RMSE, NRMSE (%), PBIAS (%), dIA, and AAE were obtained from the package of equations presented above (Equations (10)–(15)). The results for the regression coefficient b varied from 0.99 to 1.00, reaching close to 1.0, indicating that predicted and observed values were statistically similar for all crop seasons. The $r^2$ values ranged from 0.92 to 0.98, indicating that most of the total variance of the observed values was explained by the model. The RMSE values are quite low, ranging from 2.87 to 3.81 mm, indicating that the errors of estimation were small, representing values lower than 3.9% of the TAW. These values, combined with the low NRMSE (ranging from 1.78 to 2.46), indicate low residual errors. The AAE values were also quite small, ranging between 2.21 and 2.99 mm. The PBIAS were very low, indicating a slight underestimation bias in the calibration treatment (PBIAS = 0.57) and a moderate (PBIAS = 1.28) overestimation bias in DI treatment in 2019. Thus, the model did not show a trend for under or overestimation bias.

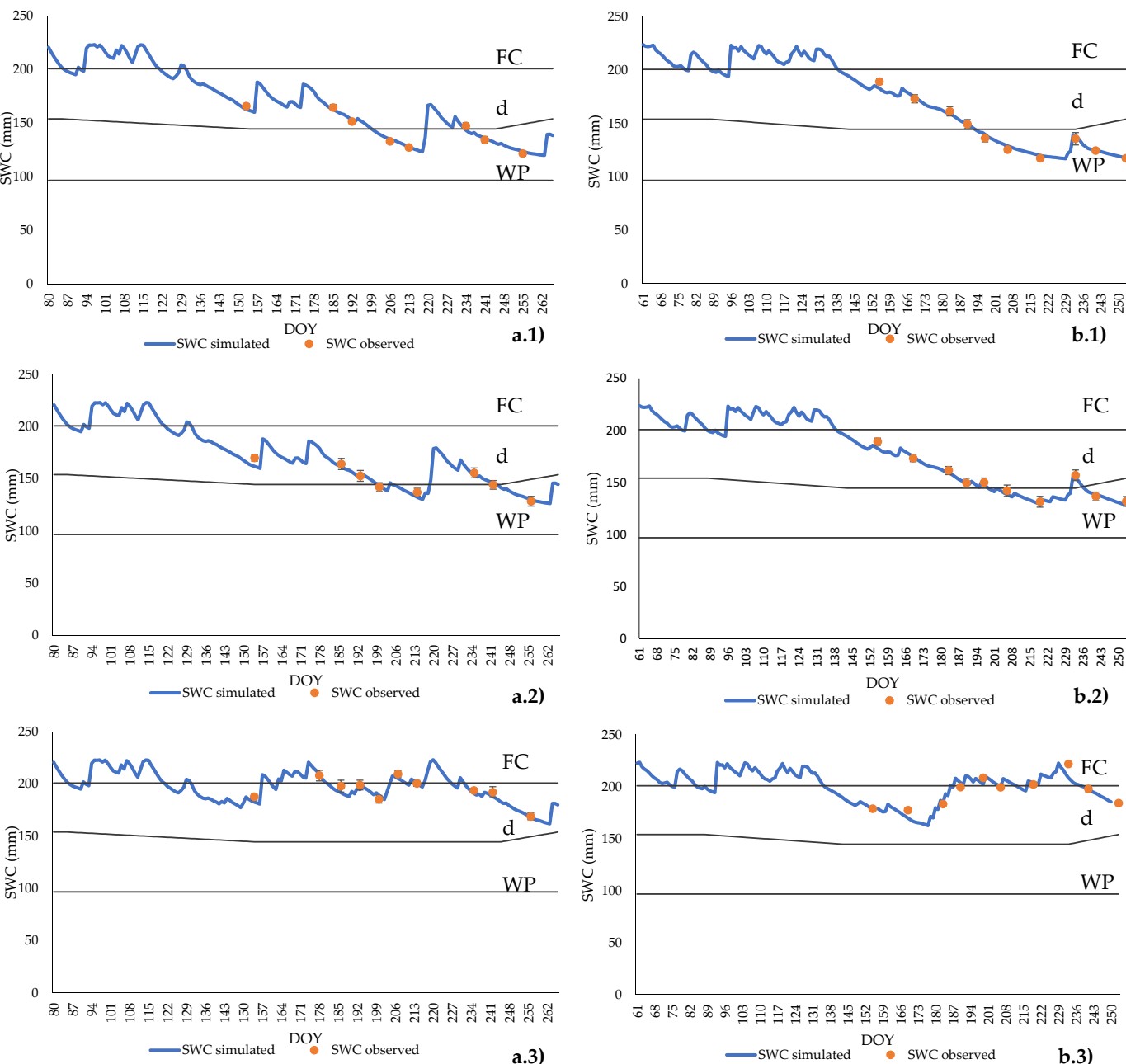

**Figure 4.** Simulated ( ———— ) vs. observed (•) average available soil water content (SWC) relative to the: (**a**) year 2019, (**b**) year 2020; (1) rainfed (R), (2) deficit irrigation (DI), (3) full irrigation (FI). DOY represents the day of the year. Error bars represent the standard deviation of the mean observed values. Curves FC, WP, and d represent the soil water content at field capacity, wilting point, and d the soil water content when depletion equals the p fraction, respectively.

**Table 5.** Goodness-of-fit indicators relative to the SIMDualKc model calibration and validations for the different treatments in the 2019 and 2020 growing seasons.

| | Year | 2019 | | | 2020 | | |
|---|---|---|---|---|---|---|---|
| | Treatment | R | DI | FI | R | DI | FI |
| Linear regression | b | 0.99 | 0.99 | 0.99 | 1.00 | 0.99 | 1.00 |
| | $r^2$ | 0.97 | 0.96 | 0.92 | 0.99 | 0.98 | 0.93 |
| Goodness-of-fit indicators | EF | 0.96 | 0.92 | 0.93 | 0.98 | 0.97 | 0.93 |
| | RMSE (mm) | 2.89 | 3.67 | 3.81 | 2.87 | 3.12 | 3.48 |
| | NRMSE (%) | 2.02 | 2.46 | 1.96 | 2.01 | 2.05 | 1.78 |
| | PBIAS (%) | 0.57 | 1.28 | 0.79 | −0.34 | 0.91 | 0.13 |
| | $d_{IA}$ | 0.99 | 0.98 | 0.98 | 1.00 | 0.99 | 0.98 |
| | AAE (mm) | 2.26 | 2.92 | 2.66 | 2.21 | 2.69 | 2.99 |

R: rainfed; DI: deficit irrigation; FI: full irrigation; b: regression coefficient, $r^2$: determination coefficient, EF: modeling efficiency; RMSE: root mean square error, NRMSE (%): normalized RMSE of total available water, PBIAS (%): percent bias of estimation; dIA: index of agreement; AAE: average absolute error (mm). Underlined data refer to calibration.

## 4. Discussion

The values obtained in this study for the HI index were higher than those observed in a study carried out with *Vitis vinifera* cv. Albariño in Rías Baixas (HI average of 1955.6 °C), Spain [46] (northwest Iberian Peninsula). This result indicates that in this region, the temperatures recorded were globally higher; this can be an indicator that distinguishe the regions. Regarding the CI index, it is possible to observe that the average minimum temperatures in September were higher in the Rías Baixas region (average of 14.7 °C) [46]. The Winkler (WI) and Huglin (HI) indices show that temperatures were lower in the year 2019 (Table 1). Associated with these lower temperatures, there was also a slightly lower annual precipitation value, and therefore, the values obtained with the application of the Seljaninov (SI) and Branas (BI) indices were very different in the two years, obtaining in 2019 a higher SI and a lower BI (Table 1). These indexes can explain the differences obtained in $K_{c\ act}$. As can be observed in Figure 3, the curve of $K_{c\ act}$ in strategy R in 2020 (warmest year) generally presents lower values when compared to the same strategy in 2019. The result of SI and BI also explains the lower SWC values in strategy R in 2020 when compared to R in 2019 (Table 1) as well as longer stress periods (Figure 4).

Regarding the standard values, the initial and midseason differences can be compared with the calibrated values. In addition to these differences in the cultural coefficients, the best fit was obtained with equal p depletion fractions, except in midseason. During the midseason, a higher value of p depletion fraction was used, which means that the midseason culture of *Vitis vinifera* cv. Loureiro is less tolerant to water stress than mentioned by the authors of [6] for the generality of *Vitis vinifera* cultivars.

Through Figure 3, it is possible to find moments in the R vineyards (Figure 3(a1,b1)), that were subject to long periods of water stress ($K_s < 1$); however, in DI vineyards (Figure 3(a2,b2)), the stress periods are lower and have also shorter duration due to the irrigations adding up to compensate water shortages. In FI vineyards, in both years (Figure 3(a3,b3)) there were no moments of water stress. Moreover, in Figure 3, it is possible to find moments, (especially at the initial growth stages) when the $K_e$ values are very similar to $K_{c\ act}$, close to 1.2. This fact is observed due to the high rainfall that occurred during these periods (Table 4) as well as the existence of low or no density and height of the soil cover vegetation. Similar trends in $K_e$ and $K_c$ values were observed by the authors of [26], to 'Godello' and 'Mencía' cultivars in Galicia, NW Spain.

Through the analysis of the results for the two seasons and three treatments (Table 4), $K_{cb\ gcover}$ and $K_{cb\ crop}$, during the mid and end seasons, were limited to their potential values due to long periods of water stress, mainly in R and DI treatments. $K_{cb\ (gcover+crop)\ act}$ during the initial stage (0.26), were lower than previous values reported by the authors of [20,47] (≈0.60). During the midseason, the values obtained for the $K_{cb\ (gcover+crop)\ act}$

(0.42) were lower than those presented by the authors of [20], which collected values of 0.65 in this period. This aspect is related to the vineyard management of the Albariño, conducted in a semi-trellised system, compared to the trellis system of the Loureiro in our study. On the contrary, in the end season, slightly higher mean values of $K_{cb\ (gcover+crop)\ act}$ are obtained for the FI treatment (0.41), being similar for the DI treatment (0.30) compared to the mean value obtained by Fandiño et al. [20] ($K_{cb\ (gcover+crop)\ act}$ = 0.30). The higher values in FI treatment are due to a greater vegetative development of the vineyard during end season, where a $K_{cb\ crop}$ was obtained (0.25), compared to the lower value, obtained by Fandiño et al. [20], with an average value of 0.14, but similar to the work of [26] with $K_{cb\ vine\ end}$ of 0.22, to 'Godello' and 'Mencía' cultivars in Galicia, NW Spain.

In relation to basal crop coefficients, similar values of $K_{cb\ crop}$ (0.18) during the midseason were obtained by Yunusa et al. [47] (0.17) to 'Sultana' vineyard with a 40% of ground cover in Australia. Other authors reported higher values for $K_{cb\ mid}$ in wine grapes: the authors of [26] report $K_{cb\ vine\ mid}$ = 0.25, the work of [48] reports $K_{cb\ mid}$ around 0.50. However, $K_{cb\ crop\ end}$ is much smaller than values reported by the authors of [48] in a 'Riesling' vineyard in the New York State, USA, who found $K_{cb\ crop\ end}$ values of 0.55.

In Figure 4, a suitable approximation between the simulated and measured SWC values can be observed, indicating that the model can predict the SWC values throughout the vineyard season using the various irrigation strategies. The values obtained by the capacitive probe demonstrate the existence of a very low error, so we can conclude that the amount of water in the soil did not vary significantly between repetitions. The $K_{cb\ full\ ini}$ (0.33) calibrated value was slightly higher to the initial period than values obtained by the authors of [20] (0.30) and relative higher to FAO-56 [6] standard values (0.20). This value is higher depending on the rainfall and active ground cover conditions. To the midseason, $K_{cb\ full\ mid}$ value (0.684) was lower than values obtained by the authors of [20] (1.15) and [26] (0.75), mainly due to active ground cover conditions and crop development. Similarly, to end season, lower values were obtained to $K_{cb\ full\ end}$ (0.54) when compared with [20] (0.90) and [26] (0.60). Final report values to $K_{cb\ full}$ were near to 'Godello' and 'Mencia' cultivars values, with similar trellis management, so that the vineyard management (trellis and active ground cover) is presented as a key factor to select the correct crop coefficients when not available to use, instead of standard values of FAO-56 [6].

The indicators obtained from the forced regression to the origin show a suitable agreement between the observed and the simulated data. The other indicators (EF, RMSE, NRMSE, PBIAS, $d_{IA}$, and AAE) demonstrate that, after a previous calibration, the model can efficiently predict the SWC throughout the crop cycle using the various irrigation strategies. The b and $r^2$ values in the present work are better than those reported by the authors of [26], who obtained b values between 0.92 and 1.04, and $r^2$ values between 0.87 and 0.97 for grapevine with active ground cover. The efficiency of the model (EF) in the present study ranged between 0.92 and 0.97, whereas the authors of [26] obtained slightly lower values between 0.77 and 0.96. The results obtained in this work are similar to those obtained with other methodologies, for example, with the results obtained by the authors of [11], who also studied vine culture. However, the accuracy of the double model approach was verified using lysimeters and with the results obtained for the cultivation of Arbequina olive [9].

Considering the totality of the indicators calculated to assess the robustness of the model, it is possible to verify that, whatever the irrigation strategy (R, DI, or FI) or year of cultivation (2019 or 2020), the values obtained by these indicators indicate a suitable fit of the model.

## 5. Conclusions

Considering the specific parameters of the study field (soil, climate, crop, and irrigation), the calibration and validation of the SIMDualKc model were successfully performed for *Vitis vinifera* cv. Loureiro. In the 2 years of the field studies, 2019 and 2020, a goodness

of fit was obtained between the SWC values observed with a capacitive probe and those simulated with the SIMDualKc model.

After making the necessary adjustments to the standard values [6], we obtained a suitable approximation between the real and simulated values. We conclude that the application of the dual approach with the SIMDualKc model is possible because, as other authors have noted, the model can correctly predict the SWC.

These values enable the determination of the SWC in future growth seasons, facilitating irrigation management by allowing farmers to adapt their management practices to achieve the levels of water stress that they want for their vineyard.

The use of calibrated parameters for cv. Loureiro allows farmers to efficiently use water. However, it is still necessary to carry out studies in the future to determine the vine response to the different levels of water stress generated in each irrigation treatment.

To facilitate the application of this calibration, it is recommended to round the values of $K_{cb\ full\ ini}$, $K_{cb\ full\ mid}$, and $K_{cb\ full\ end}$ to 0.35, 0.70, and 0.55, respectively.

**Author Contributions:** Conceptualization, M.I.V. and J.J.C.; methodology, S.P.S. and J.J.C.; software, S.P.S. and J.J.C.; validation, S.P.S., S.M., and M.I.V.; formal analysis, S.P.S. and M.I.V.; investigation, S.P.S. and M.I.V.; data curation, S.P.S., M.I.V., and C.A.-P.; writing—original draft preparation, S.P.S.; writing—review and editing, J.J.C. and M.I.V. All authors have read and agreed to the published version of the manuscript.

**Funding:** The APC was funded by Consellería de Cultura, Educación e Universidade, Xunta de Galicia (Grupos de Referencia Competitiva ED431C-2021-27).

**Institutional Review Board Statement:** Not applicable.

**Informed Consent Statement:** Not applicable.

**Acknowledgments:** This work is a result of the project TECH—Technology, Environment, Creativity and Health, Norte-01-0145-FEDER-000043, supported by Norte Portugal Regional Operational Program (NORTE 2020), under the PORTUGAL 2020 Partnership Agreement, through the European Regional Development Fund (ERDF). UIDB/05937/2020 and UIDP/05937/2020—Centre for Research and Development in Agrifood Systems and Sustainability—funded by national funds, through FCT—Fundação para a Ciência e a Tecnologia.

**Conflicts of Interest:** The authors declare no conflict of interest.

## Appendix A

In order to complement our manuscript, in relation to the state of the active ground cover, in the row and inter-row of the study vineyard, Table A1 is incorporated below.

Moreover, Figure A1 shows the evolution of density and height of active ground cover in the inter-row. The information was introduced in the SIMDualKc model to calibrate and validate the parameters required and used in the Results and Discussion sections of the manuscript. The percentages in crop row and in inter-row fraction with cover at peak canopy were from 0% to 55% for 2019 and 10% to 50% for 2020, respectively.

**Table A1.** Density and height of the green active ground cover for the 2 experimental seasons (2019–2020). Row and inter-row data.

| Year | DOY | CC_dens_Row | CC_dens_InterRow | CC_Height_Row (m) | CC_Height_intRow (m) |
|------|-----|-------------|------------------|-------------------|----------------------|
|      | 80  | 0    | 0.50 | 0    | 0.05 |
|      | 91  | 0    | 0.50 | 0    | 0.08 |
|      | 106 | 0    | 0.80 | 0    | 0.11 |
|      | 109 | 0    | 0.90 | 0    | 0.18 |
|      | 112 | 0    | 0.90 | 0    | 0.15 |
|      | 127 | 0    | 0.90 | 0    | 0.20 |
|      | 133 | 0    | 0.10 | 0    | 0.02 |
|      | 142 | 0    | 0.80 | 0    | 0.20 |
|      | 148 | 0    | 0.40 | 0    | 0.10 |
|      | 154 | 0    | 0.40 | 0    | 0.20 |
| 2019 | 162 | 0    | 0.10 | 0    | 0.10 |
|      | 166 | 0    | 0.70 | 0    | 0.15 |
|      | 178 | 0    | 0.90 | 0    | 0.15 |
|      | 186 | 0    | 0.90 | 0    | 0.25 |
|      | 193 | 0    | 0.40 | 0    | 0.08 |
|      | 200 | 0    | 0.50 | 0    | 0.12 |
|      | 207 | 0    | 0.70 | 0    | 0.15 |
|      | 214 | 0    | 0.70 | 0    | 0.15 |
|      | 235 | 0    | 0.90 | 0    | 0.25 |
|      | 248 | 0    | 0.10 | 0    | 0.05 |
|      | 266 | 0    | 0.10 | 0    | 0.05 |
|      | 61  | 0.05 | 0.80 | 0.1  | 0.25 |
|      | 75  | 0.05 | 0.80 | 0.10 | 0.30 |
|      | 132 | 0.05 | 0.80 | 0.05 | 0.10 |
|      | 137 | 0.05 | 0.20 | 0.05 | 0.05 |
|      | 143 | 0.05 | 0.50 | 0.05 | 0.10 |
|      | 156 | 0.10 | 0.75 | 0.05 | 0.15 |
|      | 170 | 0.15 | 0.85 | 0.05 | 0.20 |
| 2020 | 184 | 0.15 | 0.50 | 0.05 | 0.10 |
|      | 198 | 0.15 | 0.65 | 0.05 | 0.15 |
|      | 206 | 0.15 | 0.85 | 0.05 | 0.25 |
|      | 219 | 0.10 | 0.40 | 0.05 | 0.10 |
|      | 233 | 0.05 | 0.20 | 0.05 | 0.10 |
|      | 241 | 0.05 | 0.40 | 0.05 | 0.10 |
|      | 253 | 0.05 | 0.40 | 0.05 | 0.10 |

CC_dens_Row: density of active ground cover in the row; CC_dens_InterRow: density of active ground cover in the inter-row; CC_Height_Row: height of the active ground cover in the row (m); CC_Height_intRow: height of the active ground cover in the row (m); DOY represents the day of the year.

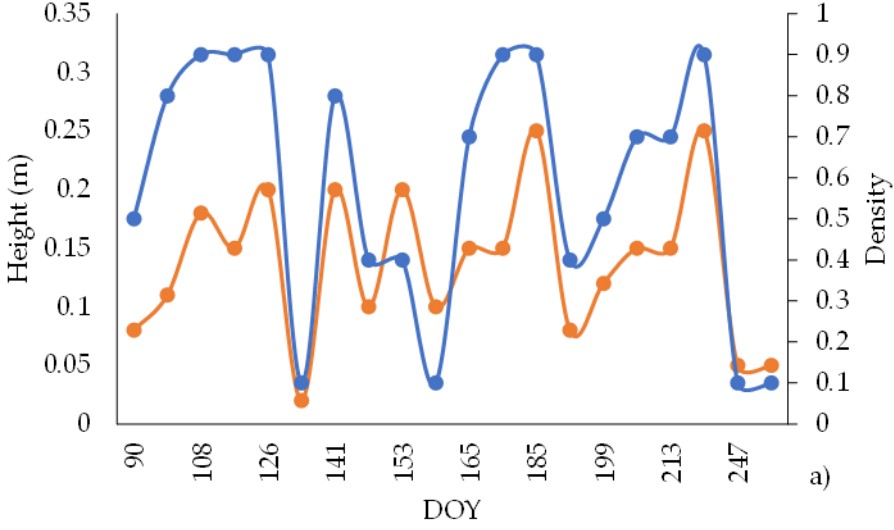

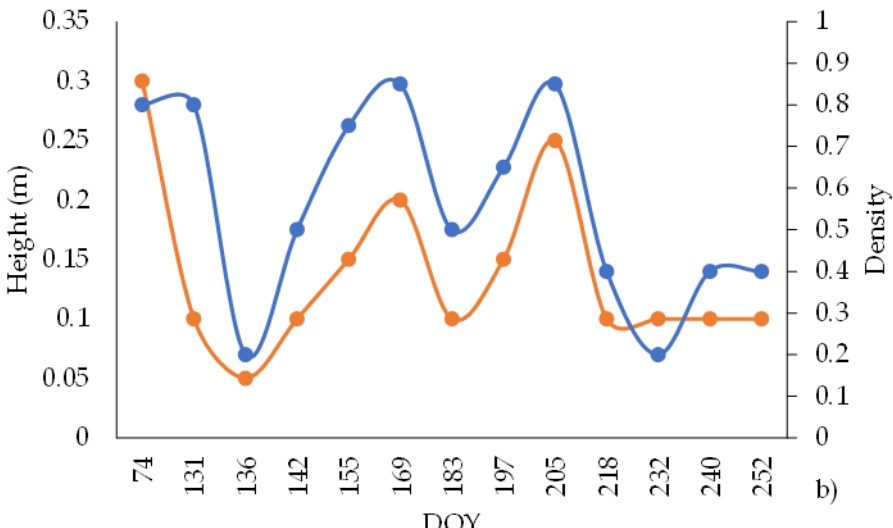

**Figure A1.** Evolution of active ground height (—●—) and density (—●—) in the vineyard inter-row relative to: (**a**) 2019, (**b**) 2020.

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
