# Peer review of "Dual Crop Coefficient Approach in Vitis vinifera L. cv. Loureiro"

_agronomy, doi:10.3390/agronomy11102062_

Round 1

Reviewer 1 Report

This Ms focus on determining crop coefficient of vine crops using SIMDualKc model. Based on comparing the soil water content measured and modeled, the adjusted crop coefficient for each growth stage is calculated. The calibrated Kc is suitable for the making local irrigation scheduling in the study region.

However, there are some drawbacks. (1) the active cover is one of the main reason that influence the Kc, adj. But authors did not analyze the effect of active cover to actual ET as well as Kc,adj. Because of the active cover, the Kc at the initial stage is approximately 1, which is much higher than that without coverage. (2) considering the low density with large plant and row spacing as well as drip line irrigation, the measured soil water content is critical for model calibration and validation. Therefore, the tube layout and how to calculate soil water content should be detailedly described. (3) based on Fig 2, Kc varied greatly across the growth season. The greater decreasing in the middle stage should be analyzed, normally, Kc is the highest during this period.

The followings are listing some comments and suggestion and others can be found in the pasted file.

  • In abstract, the Kcb gcover is not found in the text, and what's the relationship between Kcb, Kcb adj, Kcb gcover and Ke.
  • Line 43 and Eq (1), Kcb generally is defined as basal crop coefficient, that is the ratio of crop transpiration to ETo. The Kcb act defined here could confuse readers. therefore, Kcb act could be change to normal ones, like Kc,act, or just Kc.
  • Eq (2),the water balance equation is not right, check it.
  • The five index Information in Table 1 are not used in the text. Delete them.
  • In Lines 181-182 in section 2.3, The three parameters of Kcb full ini...should be defined before first use.
  • Fig 3, Y axis must be added.
  • In the discussion section, the effect of active cover fraction on Ke and Kc adj should be analyzed. this is one of the main points of this study.

Reviewer 2 Report

I think it would be more correct to use the term: "..to calibrate and validate the SIMDualKc model using the dual Kc approach" (line 78).

What was the location of the study plots, were they close to each other? What were their surfaces? 

I do not understand the context of the phrase: "to verify if cv. Loureiro is cultivated under drought stress". (line 80).

How were the bioclimatic indices values presented in Table 1 used in the further part of the study? 

How should the term "eight different soil levels" (line 139) be understood if it is stated above that the soil is homogeneous (lines 135-136). 

The article does not present the method of determining the amount of the irrigation doses (single and total), and it is not known on the basis of which the dose/s levels described as FI and DI were assumed (lines 149-157). How was irrigation controlled? For which area/s was the irrigation dose calculated? 

How was the fraction of soil shaped by the crop (Table 2) measured and how does the size compare to the active ground cover values described in lines 173-178)? Why were these elements not included in the discussion of the results. 

The publication contains few (lines number: 105/106; 107/108; 120; 126) technical errors described as: "Error! Reference source not found."

Round 2

Reviewer 1 Report

This MS now is well reviewered and can be accepted for publication.